# A Method to Qualify the Impacts of Certifications for Prefabricated Constructions

Clement Blanquet du Chayla [1,*], Pierre Blanchet [1] and Nadia Lehoux [2]

1 Wood and Forest Sciences Department, Université Laval, Quebec City, QC G1V OA6, Canada; pierre.blanchet@sbf.ulaval.ca
2 Mechanical Engineering Department, Laval University, Quebec City, QC G1V OA6, Canada; nadia.lehoux@gmc.ulaval.ca
* Correspondence: clement.duchayla@outlook.com

**Abstract:** In the province of Quebec, Canada, Small and Medium Enterprises (SMEs) in manufactured timber construction seeking to expand their market must necessarily go beyond the local trade. By exporting their products and manufactured building sections to another country, Quebec manufacturers must deal with significant regulations and certification constraints. The aim of this study is therefore to propose a method to qualify the impacts of these constraints on the export of manufactured buildings to New England in order to create a decision support tool. Since construction regulations vary depending on the location of the project, those relating to Massachusetts were analyzed, as this is currently the main destination for manufactured building sections. Considering the federal and local regulations in effect, a content analysis of the Quality Assurance Manual (QAM) set up by an industry partner and a third-party certifier enabling exports to Massachusetts was performed. In particular, the six-step method proposed by L'Écuyer was exploited for extracting and examining relevant information from regulatory texts. Through this analysis, the importance of quality control was confirmed as a keystone for certification. It also led to a better understanding of the relationships between quality control, the construction process and installation, the design and engineering choices, and the strategy to choose the project.

**Keywords:** content analysis; off-site construction; prefabricated construction; building regulations; export requirements certification

## 1. Introduction

In order to expand their market, the off-site timber-building manufacturer Small and Medium Enterprises (SMEs—fewer than 500 employees) in Quebec are seeking to expand their local Canadian market. The nearest neighbor is the United States (US), particularly New England States (NES) composed of Vermont, New Hampshire, Maine, Massachusetts, Rhode Island, and Connecticut. High exportation costs and the capacity to standardize manufactured building sections [1] encourage Quebec housing manufacturers to look for multi-residential projects in the NES. Indeed, multi-residential building projects usually offer replicable internal and external designs that suit the off-site construction philosophy.

Among the different types of manufactured buildings, 3D modules have the greatest added value but are usually more complex to build than 2D panels [2]. In addition, healthcare facilities and multi-residential buildings are considered the most likely building types for modular construction [3]. As Cid [4] highlighted, the amount of manufactured wooden buildings exported from Quebec to Massachusetts is twice the sum of all other NES exports. In order to cross the US border, every product has to be certified by a third party, which induces a production control. As construction projects are always different from each other, the impacts of certification are substantial for foreign construction industries. For these reasons, this study focuses on exports of modular construction buildings from Quebec to Massachusetts. Fully understanding the impact of the export certification during

the manufactured construction process could help to avoid flawed appraisals at the project selection stage.

The objective of this article is to create a decision support tool by characterizing the regulatory constraints to the export of manufactured timber building sections by Quebec SMEs to New England using a content analysis method. Indeed, a Quality Assurance Manual (QAM) in the context of certification of manufactured building sections is an element that has been analyzed quantitatively and qualitatively using the approach proposed by L'Écuyer [5]. It leads to the understanding and interpretation of the impacts of the export certifications on construction processes. The balance between standardization, contracts, and quality control mastery seems to be crucial to export under good conditions. The intrinsic purpose of this article is therefore to propose an example of a method for analyzing regulatory and technical documents that can be repeated in the future for similar research.

This article is divided as follows. A presentation of the different concepts explored in this study are first introduced, covering the current timber-building manufacturing industry and export regulations. The chosen methodology is next explained to understand how the study was conducted. The results and interpretations are then presented, followed by a discussion and conclusion of this article.

## 1.1. Timber-Building Manufacturing Industry

There are several levels of prefabrication that range from simple engineered wooden structures to complex modules [6,7]. These manufactured building sections can be considered products that can have different added value. Depending on the level of added value of these products, the manufacturing process can be subject to different levels of complexity. Being 3D, manufactured modules are usually more complex to produce and deliver and have the highest added value in the manufactured timber construction industry. This is why these components are considered as being the hardest construction product to export.

A manufactured building project is segmented into five stages, which are divided into different steps. RIBA [8] and Mbachu [9] both described these stages. All projects begin by a request from the customer (A) that defines the main objectives and expectations of the project. Then, the architects or designers give the preliminary design (B), which is validated by the customer. Once the global design is confirmed by engineers, clients, and architects, the details design (C) starts. The architects and engineers inform the Design for Manufacture and Assembly (DfMA) team to define every detail of the manufactured building through a Building Information Modeling (BIM) process, in order to produce, deliver, and erect it on-site. Once the details are designed and confirmed, production (D) can begin. Finally, the building sections are delivered on-site for installation (E). This framework, presented in Table 1, summarizes the distribution of activities throughout the construction process.

## 1.2. Exportation Regulations

In the US, Congress organizes the different laws within the fifty subject classifications of the United States Code (USC). Title 42 relates to Public Health and Welfare, which includes a chapter on Manufactured Home Construction and Safety Standards [10]. The Code of Federal Regulations (CFR) is the interpretation of the USC by the relevant department. The CFR includes more information on statutory interpretation, making its technical application easier than the USC. Title 24 of the CFR contains the Manufactured Home Regulations [11], providing a framework for the regulations followed by the US states. States may then add local regulations and statutes that are consistent with the CFR and USC [12]. In Massachusetts, the Code of Massachusetts Regulations [13] called Manufactured buildings, building components, and mobile homes defines the local rules.

**Table 1.** Prefabricated construction project sequence with task descriptions.

| Category | A | B | C | D | E |
|---|---|---|---|---|---|
| Number | Customer's Request | Preliminary Design | Detail Design | Production | Installation |
| 1 | Discussions on customer expectations | First plans to identify the characteristics of the building with the client | Design Detail | Foundations and services | Transport and installation of the units |
| 2 | Getting information from the site | Complete brief of rooms, layouts, interior and exterior specifications, construction process | Agreements on interior and exterior specifications | Ordering materials | Finishes |
| 3 | Indicative budget | Preliminary budget forecast | Structural Engineers Report | Follow-up of annual Health and Safety plans | Passing access to units |
| 4 | Delivery time | Provisional program | Mechanical, electrical and plumbing design | Quality control checklist | Donation of operation and maintenance documents |
| 5 | Risk Assessment | Complete and frozen design | Integrated design | Final logistics plan | |
| 6 | | Full cost assessment and agreement | Freezes specifications for interiors, exteriors, permanent fixtures and furnishings | Acceptance of finalization of the unit in factory | |
| 7 | | Building Permit | Drawings for manufacturing | Quality control | |
| 8 | | Client's signature | Preparation of quality control checklists | | |
| 9 | | | Ordering materials | | |
| 10 | | | Preparation of logistics / lifting plan | | |
| 11 | | | Search for third party quality control agencies | | |
| 12 | | | Issuance of all documents required for certifications | | |

The CFR Subpart E Manufacturer Inspection and Certification Requirements defines the third-party corporate certification (CFR 24:5.1.4.1.3.5—SUBPART E), which is divided into two sections: the Design Approval Primary Inspection Agencies (DAPIA) and the Production Inspection Primary Inspection Agencies (IPIA).

### 1.2.1. DAPIA

The DAPIA ensures that the design respects local regulations and technical requirements. They also approve the quality control program through the Quality Assurance Manual (QAM).

### 1.2.2. IPIA

The IPIA complements the DAPIA by verifying that the QAM and technical construction details are applied and followed at the production stage. If the agency assesses that the manufacturer is performing adequately, a Certification Report is issued. In addition, the IPIA monitors the production process to ensure compliance with the design specifications and the QAM. Once the manufactured module is completed and the desired standards are satisfied, a permanent label is provided and affixed by the IPIA and the manufacturer places a Data Plate. The Data Plate contains the manufacturer and main contractor details as well as the main technical details of the project. It is installed in a secure and accessible location in each module. On the construction site, the building official is responsible for ensuring that the building conforms to the construction drawings and planned processes as approved by the DAPIA. This is a local government representative. In addition, the role of this official is to verify that Occupational Safety and Health Administration (OSHA) requirements are respected on-site [14].

### 1.2.3. Technical Standards

Most of the technical construction standards in the U.S. are defined by the International Code Council (ICC). Several codes exist, such as: International Building Code (IBC), International Residential Code (IRC), International Mechanical Code (IMC), International Plumbing Code (IPC), International Energy Conservation Code (IECC), National Electrical Code (NEC), etc. These codes are revised every three years and each state decides which edition to follow. Some states or cities may even decide to follow their own standards and building code, which are usually complementary to the existing ICC standard. These alternative or complementary standards usually refer to local urbanism norms and transportation regulations. ICC standards outline construction specifications and requirements and, in accordance with these, every building component must comply and be certified by an adequate third party, such as the International Code Council Evaluation Service (ICC-ES) [15] or Underwriters Laboratories (UL) [16].

### 1.2.4. Process Map Summary

To visualize the certification flow, Figure 1 summarizes the interaction between the US regulations and certifications with the various project stages of a manufactured construction process.

This section is divided into two parts. To lead to the decision of the chosen content analysis approach, the Analysis Methods Literature Review part outlines the considerations and matters of the analysis method from the perspective of communication of results. Then, the Detailed Method part allows us to deepen the chosen method and to explain the actions undertaken.

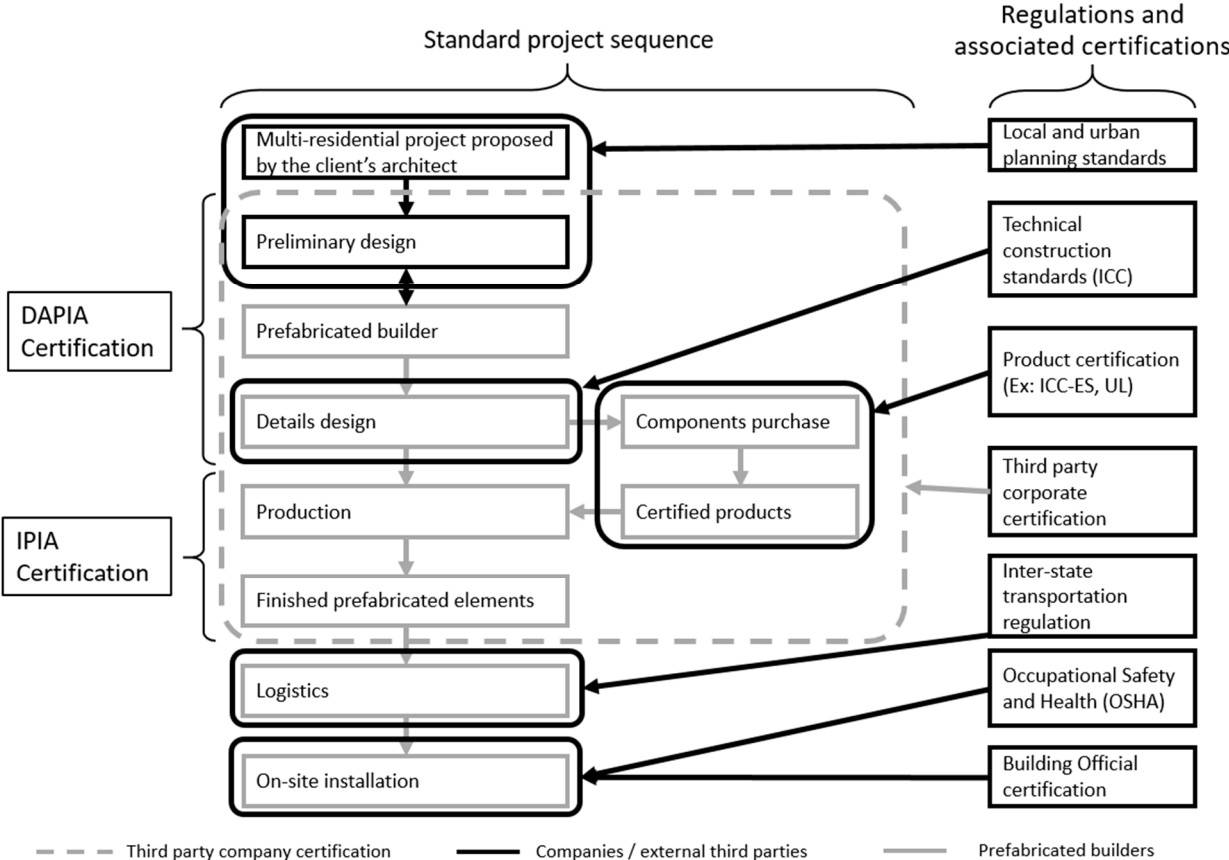

**Figure 1.** Certifications and regulations process map summary in the building construction industry in US. 2. Materials and Methods.

### 1.3. Analysis Methods Literature Review

Content analysis is a qualitative research method that uses a series of procedures to make valid inferences from a given text. This analysis can be done using a quantitative or qualitative approach. The quantitative approach creates and tests a coding system [17]. The critical and most difficult part of the analysis is to define mutually exclusive and sufficiently broad categories. The Lasswell Value Dictionary and Harvard VI Dictionary are examples of categories that can be used to analyze a text. This analysis method usually requires computer assistance, which can be a source of issues. The qualitative approach can be used for both written text and verbal transcription. However, as with the quantitative approach, words or sentences can be interpreted in different ways.

Content classification [18] can be seen as monothetic—categories containing cases that are exactly the same on all measured variables or dimensions—or polythetic—categories representing groups of cases with broad general similarities or common characteristics. Classification is a descriptive tool that aims to reduce complexity and identify similarities and differences. However, classification is descriptive or pre-explanatory, and can suffer from a lack of usability.

Mixing these content analysis approaches, L'Écuyer proposed a six-step methodology. L'Écuyer's method is a broad and general approach that draws on a literature review of qualitative analysis methods. The following different steps aim to reduce bias and lack of objectivity. First, this method proposes to read the collected material several times and to divide the contents into smaller datasets. Then, the categorization of the information needs to be done, which consists of grouping statements that have a similar meaning. A category should be a common denominator into which a set of statements can be naturally incorporated. Following these primary stages, the categories are quantified in terms of frequencies, percentages, or various other indicators. It allows the extraction of a scientific

description based on quantitative and qualitative analysis, often used to explain the results of the quantitative analysis. The final stage is to interpret the results. L'Écuyer's method has been used in similar research context [19,20].

Following the content analysis, the results must be clearly interpreted. Studies indicate that presenting information in graphs has a greater impact on participants' understanding [21]. Moreover, small paragraphs and bullet points can be used to facilitate understanding and highlight important elements. Another technique for communicating complex information can be the use of trees and maps [22]. Trees are, by definition, rooted, which means that there is an origin to the succession of information [23]. A map is when there is more than one linearity (when there are few 'roots'), even if the information is chronological. For example, Figure 1 is considered as a process map, although it has an origin (multi-residential project proposed by the client's architects). The nodes/links allow users to establish relationships/states between information. This allows a simple understanding of a complex and multi-criteria problem [24]. Trees and maps can be descriptive and factual (composition, attribute, process, etc.) or conceptual to convey an idea or a thought path. The final results and their interpretation will be communicated based on these principles in order to propose a method to successfully simplify and understand the different interactions of the proposed inductions.

### 1.4. Method Description

As a reminder, the objective of this article is to characterize the regulatory constraints in the export of manufactured timber-building sections from Quebec companies to New England. L'Écuyer's method was used to analyze the technical content needed so as to create a decision support tool. L'Écuyer's method was selected as it focuses on identifying the specific characteristics of the themes analyzed so as to extract relevant trends or patterns. As a result, the critical meaning of the concepts analyzed comes from the specific nature of the content studied. As highlighted in Figure 1, the technical requirements that are reviewed by the DAPIA are directly subjected to ICC standards. Indeed, maintaining smooth relations with the DAPIA and successful certifications depend solely on the experience and building code knowledge of the manufactured construction company's designers and engineers. Despite the fact that different codes are followed depending on the project location, the technical details and local regulations are considered static information, as they do not change often.

However, the IPIA uses dynamic information. Every company is different, composed of people with different responsibilities, skills, capacities, equipment, etc. In addition, every construction project is different and requires particular building module designs, even with a construction method that tends to be as standardized as possible. These company-related and project-related diversity factors are uncontrollable unless the QAM is well done and complete. Indeed, the QAM controls technical specifications, production, certification, and human resources and responsibility through the quality control framework. This manual is the keystone of the certification process because it controls the qualitative aspect and the part of the project most prone to disorganization.

As a case study, a modular construction industry partner of the research was studied. This company exports modular houses to the US and is accredited by PFS Corp in New England, which sees to both the DAPIA and IPIA. In particular, since the accreditation process is state-driven, the QAM used specifically by the company for exports to Massachusetts was analyzed to better highlight the impact of this process. A content analysis of this manual was carried out using L'Écuyer's (1987) step-by-step method:

**Step 1:** Several readings of the QAM confirmed that the document describes:

- DAPIA's scope of work: Verification of the construction process, plans, technical details, and traceability and conformity of materials used.
- IPIA's scope of work: Verification of quality controls at each stage of production, traceability of issues and discrepancies, certification of problem-solving during the production stage, and allocation of responsibilities within the company.

In addition, it is interesting to highlight that there is no specific mention of manufactured modules in the QAM. The QAM is written for the production line of the company, in accordance with their construction methods and internal organization. As long as the final product follows the standard certification and is validated by DAPIA requirements, the company can build any type of building section.

**Step 2:** The QAM is already partitioned into sections and subsections that divide the content into smaller datasets. As the QAM analyzed contained 61 pages of the company's internal confidential information, requests for further details may be addressed to the authors of this article.

**Step 3:** The well-known 5 Ws (What, Who, When, Where, and Why) supported the elaboration of categories. The What was determined by understanding whether the content analyzed was characterized by information that the company should give or instructions that should be followed. The Who is related to the responsibility that the QAM gives to the different trades in the company. The Where and When were determined by identifying the allocation of instruction or information in the construction process. The Why corresponds to the reason behind this analysis, in order to crystallize the important information to determine the impacts of certification. To summarize, the chosen categorizations are as follows:

- Information or instruction?
- What trade?
- In the project, where and when it relates to?

**Step 4:** Having content divided into sections and subsections helped to quantify the categories. It allowed us to identify the instructions of the QAM and the information given by the company, as well as determining the predominant character of the content.

The number of trades mentioned, regardless of section allocation, was counted to identify the distribution of responsibilities given by the QAM. Considering the size of the content, counting was done manually. Finally, all sections and subsections were allocated to one or more stages of the construction process (defined in Table 1).

**Step 5:** The scientific description, based on the quantitative and qualitative analysis, is presented graphically in the following section.

**Step 6:** The interpretation of the results is detailed in the following section.

## 2. Results

This section presents the results obtained from analysis of the QAM content of a timber-building manufacturer SME based on L'Écuyer's method.

The first category proposed in step 3 of the analysis was whether a specific section or subsection of the QAM should be classified as instruction or information. An instruction, in this study, is considered as the act of providing with authoritative directions, while information is the knowledge communicated or received about a particular fact or circumstance. Figure 2 presents the data character distribution results. It demonstrates that instructions are predominant in the QAM. However, the information related to the company identity is also essential for the QAM issue.

The second category related to the trades targeted by each section or subsection of the QAM, in order to identify the distribution of responsibilities. The results, given in Figure 3, highlight that the Quality Control Manager and Quality Control Inspector are the most frequently mentioned, unlike the other trades. This confirms that the QAM's first priority is to supervise and oversee (mostly through instructions) this quality control work, which allows issues during the manufacturing process to be reduced. However, the results also highlight the importance of the Design and Engineering (D&E) Manager, who comes in third, followed by the Procurement Manager. Indeed, these two trades are controlled by the DAPIA before production begins, but they still have a substantial impact on—and during—the quality control process.

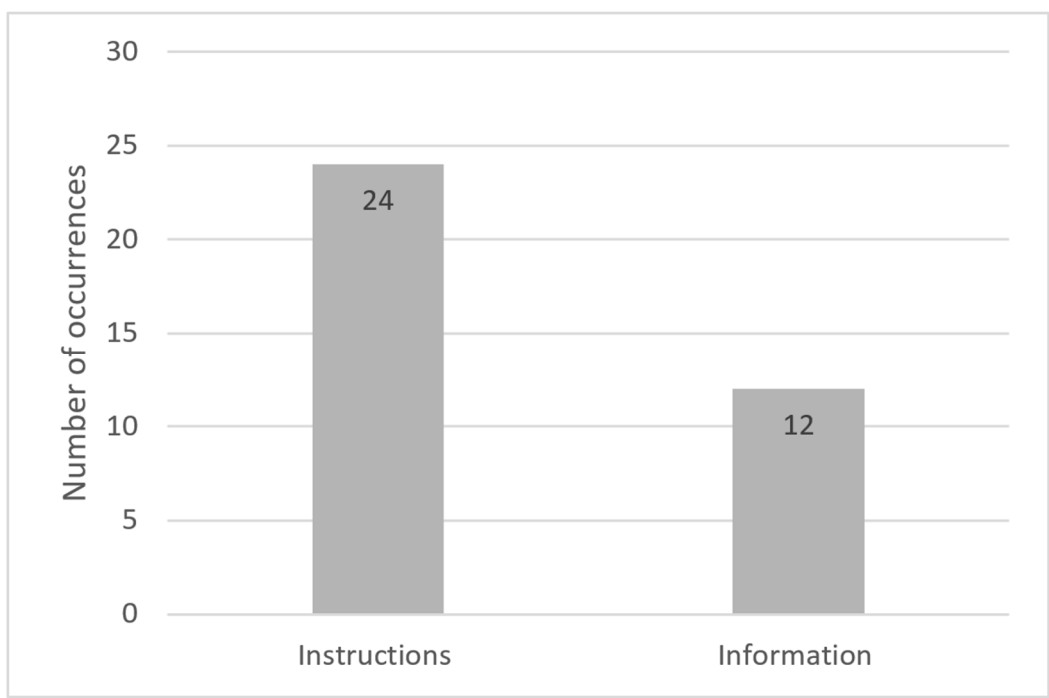

**Figure 2.** Data character of the Quality Assurance Manual for exportation certification to Massachusetts.

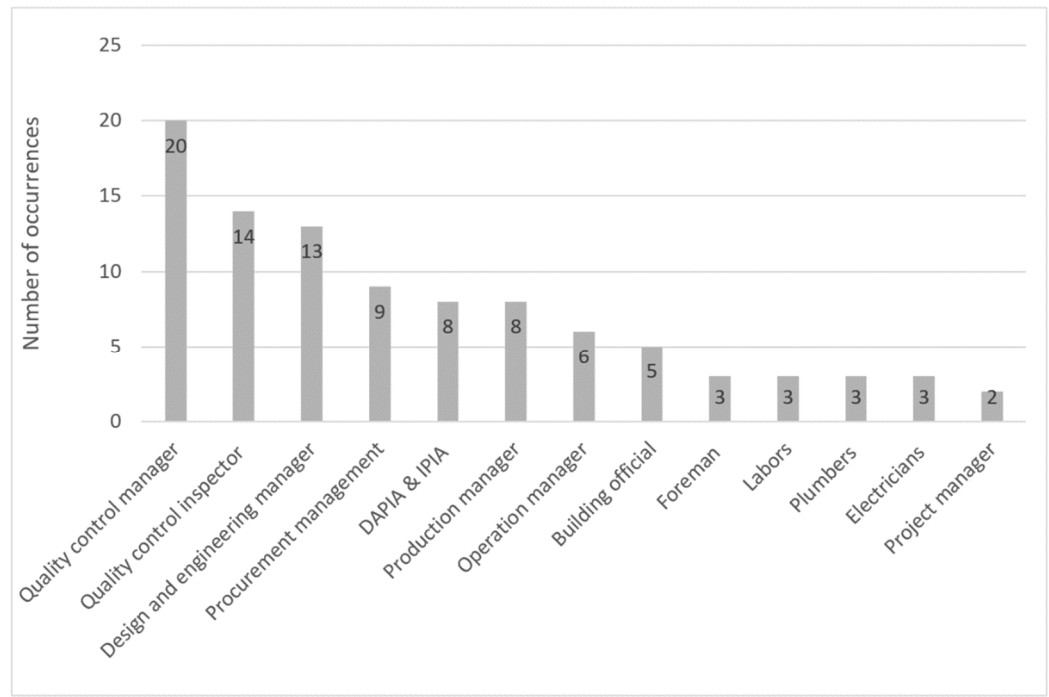

**Figure 3.** Responsibility of trades allocated by the Quality Assurance Manual for exportation certification to Massachusetts.

Finally, all sections and subsections were allocated to one or more stages of the construction process. The results, presented in Figure 4, show that the QAM focuses slightly more on the production stage than on-site installation. However, it confirms that the QAM concentrates its instructions and information on the handling part of the project, with the design parts mentioned as references.

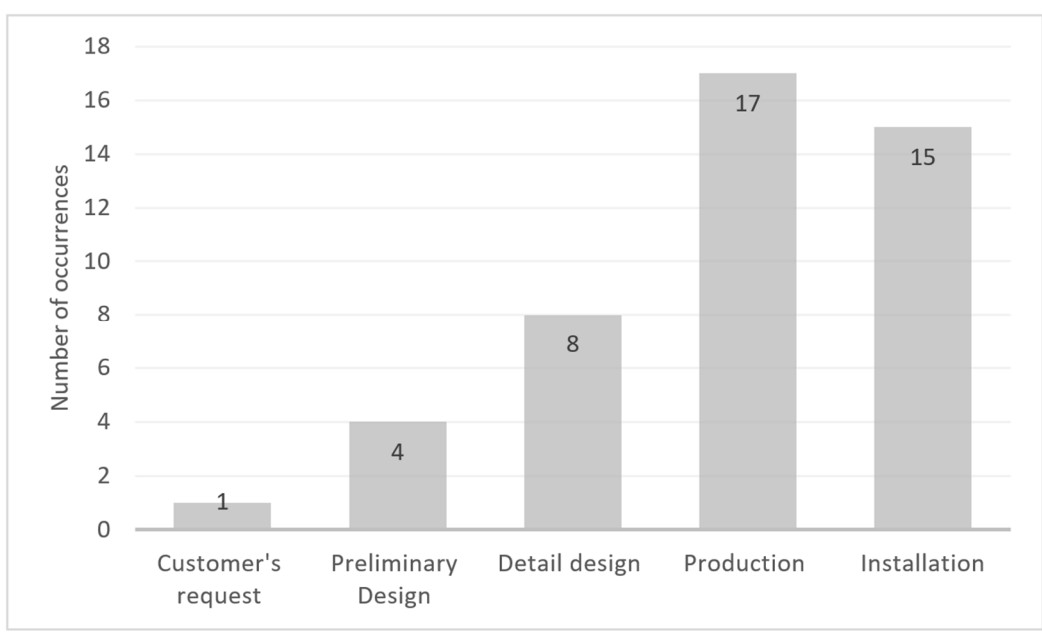

**Figure 4.** Main project stages targeted in the Quality Assurance Manual for exportation certification to Massachusetts.

Considering these quantitative results, it seems clear that the quality control teams (manager and investigators) play an important role in the production and installation stages. The transdisciplinary role of the Quality Control team is a crucial element of manufacturer and product accreditation, and is beneficial to all stakeholders: customers, manufacturers, and local authorities. As outlined in the responsibilities of the Quality Control manager, the person in charge of this trade must have enough design, engineering, social, and leadership skills. However, D&E managers and teams also play a major role as they influence the efficiency and success of quality control, certification, production, and installation. Therefore, a major part of the profitability of the project to be exported is based on the D&E work.

*Information Interpretation*

As the D&E work is crucial to the project's profitability and successful export certification, choosing the right construction project seems to be very important for SMEs. Production capacity, customer lead-time, and costs are the usual criteria, but for accreditation, the level of standardization and prefabrication becomes decisive. For the Quality Control team, the more the product is standardized, the easier it is to inspect it. The process of mass standardization principle is to replicate the same product (or project) in order to produce it more efficiently and, therefore, with more profitability. The mass standardization process in construction is usually used in hotels, offices, or multi-residential housing. However, the client usually wants its own single and customized home. Mass customization could then be used by SMEs for exported projects as it aims to avoid full customization by pre-empting purchaser choice, ensuring that there is a wide range of design types [25]. Using flexible standardized build-ups and multi-performance materials [26] can also simplify the construction process, design details, and installation.

In addition, knowing export constraints on the technical, logistical, and legal levels, the manufacturer's contractual position in the project needs to be well defined. Indeed, the manufacturer could simply be a contractor involved only in the production stage of a

project. However, the manufacturer could also be involved in the design stage or even be the project manager.

Figure 5 is a conceptual graph that illustrates that the certification process is simplified with increasing efficiency and transparency of the quality control, construction, and installation processes. The shape of the curves may depend on the responsibilities of the manufacturer in the project and its construction process. As this conceptual example shows, the accreditation can be more easily achieved with a project using mass standardization (A). Mass customization (B) is an alternative to pure standardization (A) or customized projects (C). Mass customization (B) gives an in-between that allows the project to be customized by the client using known internal construction processes and material choices. These allow access to a wider range of projects while keeping the export accreditation process rather simple, particularly for multi-residential projects. In most cases, the more control the manufacturer has over its environment and products—in other words, the closer it is to (D) on the X-axis—the easier it is for the company to export a wider variety of projects.

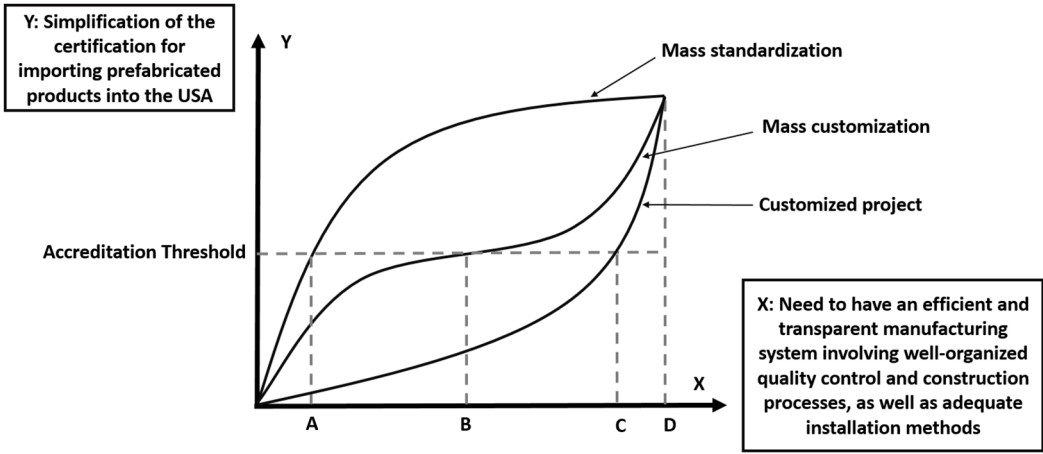

**Figure 5.** Conceptual representation of the accreditation process for exportation certification to US.

The architectural complexity of a given project to be exported will lead the D&E and quality control teams to face technical, logistical, and legal issues that can induce accreditation barriers. Figure 6 presents a decision-making tool to support companies in evaluating the risk and profitability of a project according to its architectural complexity, the manufacturer's role in the project, and its control over key accreditation processes (quality control, construction, and installation).

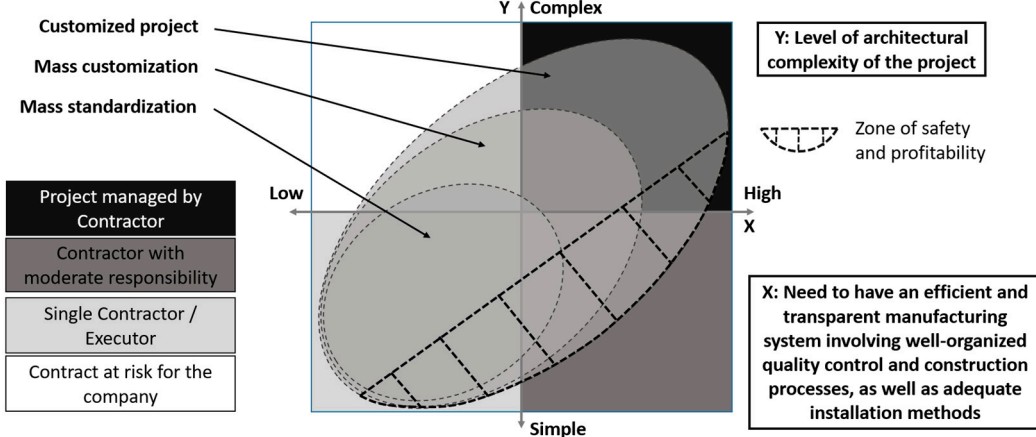

**Figure 6.** Conceptual 4-quadrant graph—exported project profitability analysis tool.

The three areas determining mass standardization, mass customization, and customized projects are mutually inclusive, and show the scope of action that these methods can reach. Mass standardization represents a low level of architectural complexity because it is a repetitive method. A more customized method has a wider scope.

The background shades of grey define the responsibilities of the manufacturer. Indeed, a contract may entail many different levels of responsibilities, but here, three distinct levels have been defined. A Single Contractor would simply issue a purchase order without implementing any D&E. A Project managed by the contractor requires them to lead the project from beginning to end, taking on the heavy responsibilities. The Contractor with moderate responsibility is in-between, which is the usual case in manufactured construction. As mentioned above, low production control coupled with high architectural complexity is an area of risk for the manufacturer. More precisely, a contract could be seen as at risk when a project involves a significant complexity level from the point of view of the system architecture and reduced control over the manufacturing processes. In such a case, the company may have difficulties in managing operations to meet expectations as well as to track and report the multiple problems faced. This situation makes it more difficult (or even impossible in a reasonable price range) for the DAPIA and IPIA to grant exportation documents and certify a QAM. A manufacturer with a Single Contractor role, with a little or moderate control over its production process, will position itself more towards architecturally simple projects to facilitate its accreditation. As the manufacturer's efficiency and transparency increases, so does its responsibility within the project, including the company's possibility of becoming a project manager with a high level of architectural complexity.

Mass customization and customized projects gradually and successively increase the architectural complexity that can be achieved as well as the responsibilities of the manufacturer. However, it also increases the risk area if the minimum level of production control is not adequate.

### 3. Discussion

Figure 6 illustrates the problem faced by the manufactured construction SMEs in Quebec currently seeking to export or for other Canadian provinces. The level of quality control in the production and installation processes is high compared to the design flexibility expected by the market. Current business volume (and demand) in New England is not specified on this graph, but as the black dotted Zone of Safety and Profitability line shows, projects undertaken must be as far as possible from the risk area. The key performance that a project should aim for is to ensure that the project stays in a safe and profitable zone. The balance between standardization, contracts, and quality control seems crucial to exporting in the best conditions.

Therefore, manufactured construction SMEs initiating their exports should look for projects with repetitive and standardized modules, such as basic hotels or multi-residential building architectures, to ensure that their building process will not be an additional barrier to the complexity of the certification. With more experience, SMEs will benefit from exportation skilled employees and construction processes and will be able to take on more risks, with projects of higher architectural complexity and perhaps better profitability.

Although this study was looking at the export of manufactured timber-building sections from SMEs in Quebec to Massachusetts, it is worth mentioning that the export process will be similar for a larger company or for a different type of prefabricated construction (e.g., panelized construction, prefabricated steel framed construction, etc.). The regulations imposed on the manufacturer and certified by the DAPIA and IPIA will also be those applicable in the erection site area. Therefore, the methodology proposed in this research could be used and adapted to conduct similar studies on the export of prefabricated construction in general to the US.

## 4. Conclusions

The purpose of this study was to assess the impacts of export certification of manufactured building sections to support Quebec construction SMEs seeking to expand their commercial market. To illustrate the complexity of this accreditation process, the QAM used by a Quebec SME for export to Massachusetts was analyzed using L'Écuyer's (1987) content analysis method.

The content analysis focused on identifying three characteristics for each section or subsection of the QAM: (i) the data's character (information or instruction), (ii) the trade involved, and (iii) the stage of the construction process. The results highlighted that the document contains twice as much instruction for the company as information. The trades most involved in certification are the Quality Control and the D&E teams. Finally, the QAM predominantly regulates the production and installation steps of the project. From these results, conceptual graphs were proposed as tools to support SMEs in identifying the feasibility of exporting manufactured building projects based on the level of customization and company responsibilities.

The conceptual graphs used to qualify the impacts of certification for manufactured construction exportation revealed the importance of visualization, especially when it comes to interpreting several criteria simultaneously. The level of standardization, company responsibilities, and quality control processes used seem to define the level of difficulty in exporting across the US border. The more standardized the manufactured sections, the easier it is to control the quality of the products and the building process, which facilitates exportation. For their initial export projects, manufactured construction SMEs should look for simple and repeatable architectures as well as minimal company responsibilities to reduce the profitability risk. As the exportation knowledge and skills of the D&E, Quality Control, and Production teams increases, the architectural complexity of projects and the level of company responsibility may also increase.

The tool for analyzing the profitability of exported projects proposed in Figure 6 could be further developed to use measurable indicators to help SMEs to position export projects on the graph. In addition, L'Écuyer's (1987) method could be used for similar research to analyze certification or regulatory content qualitatively and quantitatively.

**Author Contributions:** C.B.d.C.: conceptualization, methodology, formal analysis, investigation, and writing—original draft. P.B.: methodology, supervision, validation, and writing—review and editing. N.L.: methodology, supervision, and writing—review and editing. All authors have read and agreed to the published version of the manuscript.

**Funding:** The authors are grateful to Natural Sciences and Engineering Research Council of Canada for the financial support through its IRC and CRD programs (IRCPJ 461745-18 and RDCPJ 514294-17) as well as the industrial partners of the NSERC industrial chair on eco-responsible wood construction (CIRCERB), the industrial partners of the industrialized construction initiative (ICI) and the Créneau Accord Bois Chaudière-Appalaches (BOCA).

**Institutional Review Board Statement:** Not applicable.

**Informed Consent Statement:** Not applicable.

**Data Availability Statement:** The data presented in this study are available upon request to the corresponding author.

**Acknowledgments:** The authors are grateful to ProFab, MaisonLaprise, and Ultratec for the materials used for the technical support given.

**Conflicts of Interest:** The authors declare that they have no known competing financial interests or personal relationships that could have appeared to influence the work reported in this paper.

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
