# Peer review of "A Method to Qualify the Impacts of Certifications for Prefabricated Constructions"

_buildings, doi:10.3390/buildings11080331_

Round 1
Reviewer 1 Report
Comments for the author.
Introduction is prepared well, systematically and in a high understandable level. Methodology and results are well presented and organized. Conclusions are exhaustive and supported by the results. The presented content of the paper offers the original research results.
The article deals with the issue of export certification for wooden structures manufactured in Canada. When exporting their products, manufacturers of these elements have to face significant regulations that limit exports. The article attempts to create a tool supporting decision-making regarding the target export of products. The research was based on the current export of items to the state of Massachusetts. As a result, factors were identified and conceptual graphs were proposed to support the definition of export opportunities. The strong point of the article is the great versatility of the developed method.
Notes:
1.
Overall language quality is good in my opinion.
2.
Abstract should include a sentence or two about the used content analysis method.
3.
Figure 1 is a table 1, not a figure 1.
3.
All figures and tables have no source.
4.
Figure 1 (Table 1) is hardly legible.
5.
Figures 3, 4, 5, nie mają jednostek.
Author Response
The authors would like to thank the reviewers for their constructive comments that helped us improve the manuscript. The article has been reviewed and modified so that all comments pointed-out by the referees are clarified and/or implemented. Following are our responses to the comments received from the reviewers.
Comments from Reviewer 1
Comment 1: Abstract should include a sentence or two about the used content analysis method.
Response: As suggested, the abstract now specifies that the six-step method proposed by L’Écuyer was the one exploited to extract and examine relevant information from regulatory texts.
Comment 2: Figure 1 is a table 1, not a figure 1.
Response: The title for the first table has been modified.
Comment 3: Figure 1 (Table 1) is hardly legible.
Response: In order to facilitate the reading, the text font of the table has been modified.
Comment 4: Figures 3, 4, 5, do not have units.
Response: In fact, the y axis for Figures 3, 4, and 5 refers to the number of occurrences. To avoid any misunderstandings, the title of the y axis has been added in each figure in the new version of the article.
Reviewer 2 Report
The authors introduce a decision support tool to understand the impact of regulatory constraints in the export of timber-building sections. The paper does have certain research contributions, but it also shows a few points to revise further, and thus, the reviewer recommends a minor revision.
- The authors described the L’Ecuyer’s method selected as the basis of this study, but more explanations and justifications on why this method is superior to other available methods should be discussed further.
- What were assumptions that the authors made to use the L’Ecuyer’s method to analyzing the QAM content of a timber-building manufacturere SME?
- How can the results from the case study be generalized to other projects or other types of prefabricated components other than timber-building components? What are some of the factors that would help increase the generality of the developed methodology?
Author Response
The authors would like to thank the reviewers for their constructive comments that helped us improve the manuscript. The article has been reviewed and modified so that all comments pointed-out by the referees are clarified and/or implemented. Following are our responses to the comments received from the reviewers.
Comments from Reviewer 2
Comment 1: The authors described the L’Ecuyer’s method selected as the basis of this study, but more explanations and justifications on why this method is superior to other available methods should be discussed further.
Response: As highlighted by the reviewer, different methods have been proposed over the years to examine texts, identify recurring concepts, and interpret them adequately (Nassaji, 2015). As an example, Bengtsson (2016) suggested four stages to conduct a qualitative content analysis, involving decontextualization, recontextualization, categorisation, and compilation. L’Écuyer’s method was the one selected as it focuses on identifying the specific characteristics of the themes analyzed so as to extract relevant trends or patterns. As a result, the critical meaning of the concepts analyzed comes from the specific nature of the contents studied. The method has to be conducted based on six steps, which are reading several times, dividing the content, categorizing information, quantifying the categories, extracting the scientific knowledge, and interpreting the results. The research team has furthermore already used the method in past research and the results generated were each time especially relevant.
Some details have been added in the method description section to better justify the choice made in the updated paper.
Comment 2: What were assumptions that the authors made to use the L’Ecuyer’s method to analyzing the QAM content of a timber-building manufacturere SME?
Response: L'Écuyer's method is a broad and general approach that draws on a literature review of qualitative analysis methods. The different steps of the method aim to reduce bias and lack of objectivity.
To apply the method, no assumptions are necessary.
Changes have been made in the article to clarify this point.
Comment 3: How can the results from the case study be generalized to other projects or other types of prefabricated components other than timber-building components? What are some of the factors that would help increase the generality of the developed methodology?
Response: Although this study was looking at the export of manufactured timber building sections from SMEs in Quebec to Massachusetts, it is worth mentioning that the export process will be similar for a larger company or for a different type of prefabricated construction (e.g., panelized construction, prefabricated steel framed construction, etc.). The regulations imposed on the manufacturer and certified by DAPIA and IPIA will also be those applicable in the erection site area. Therefore, the methodology proposed in this research could certainly be used and adapted to conduct similar studies on the export of prefabricated construction in general to the USA.
This information has been added in the discussion section in the new version of the article.
Reviewer 3 Report
International project seems to grab importance as considering the optimization of carbon burden, for each country. In this regard, the paper describes a trial example. Several comments are given to improve the contents of the paper.
On line 359, Figure 6 is typo. In Fig.7, white zone is expressed as contractor at risk for the company. Explain what kind of risks there could be. In addition, the next zone is the black zone where is expressed the zone project manages deals with. Generally, the more the complexity level is, the higher/more risks there are. This is not well investigated in the paper.
In addition, there are several companies, Small and Medium Enterprises, Project management, Construction, Design/Engineering. Is it always true that all projects are handled by an independent project manager ?. It is supposed that a general contraction company can handle all the roles. Explain which project type is popular (a project manager type, or a general contractor type) in Quebec and Massachusetts.
Author Response
The authors would like to thank the reviewers for their constructive comments that helped us improve the manuscript. The article has been reviewed and modified so that all comments pointed-out by the referees are clarified and/or implemented. Following are our responses to the comments received from the reviewers.
Comments from Reviewer 3
Comment 1: On line 359, Figure 6 is typo.
Response: As suggested, the figures’ numbers have been corrected.
Comment 2: In Fig.7, white zone is expressed as contractor at risk for the company. Explain what kind of risks there could be.
Response: In fact, a contract can be interpreted as “at risk” when a project involves a significant complexity level from the point of view of the system architecture and a reduced control over the manufacturing processes. In such a case, the company may have difficulties to manage operations to meet expectations as well as to track and report the multiple problems faced. The DAPIA and IPIA could then lose faith and decide to not certify the company.
Some details concerning this remark have been added in the Information interpretation section.
Comment 3: In addition, the next zone is the black zone where is expressed the zone project manages deals with. Generally, the more the complexity level is, the higher/more risks there are. This is not well investigated in the paper.
Response: In a construction project, a manufactured construction company can have different responsibilities and, as a result, assume different levels of risk. In order to share responsibility and risk or fulfil the project requirements, the company may use other contractors for the design part, the project management part, and even the engineering part, depending on the skills they are looking for.
The black area in Figure 7 (now Figure 5) has been re-defined as “Project managed by the contractor” in the updated article to highlight the fact that the manufactured construction company can deal with different levels of responsibility while avoiding any misunderstandings.
Comment 4: 1) In addition, there are several companies, Small and Medium Enterprises, Project management, Construction, Design/Engineering. Is it always true that all projects are handled by an independent project manager? It is supposed that a general contraction company can handle all the roles. Explain which project type is popular (a project manager type, or a general contractor type) in Quebec and Massachusetts.
Response: Most of the manufactured construction companies in Canada in general and in Quebec in particular are considered as small and medium enterprises (ISED, 2019). These companies are typically responsible for conducting DfMA (Design for Manufacture and Assembly), following the Project manager and Design & Engineering requirements. This remains true no matter who is responsible for the Project manager aspect or the Design & Engineering part (i.e., in or out the manufactured construction company).
As mentioned previously, the conceptual representation has been modified to avoid any confusion.
For the following questions, according to our research no study gives precise indications on these subjects yet, we can only give you answers based on our appreciation and experience of the industrial ecosystem.
No, sometimes they are not handled by an independent project manager, each stakeholder could play this role (it often depends on the size of the project). Often it is the property developer who is General Contractor and Project Manager. Changes have been made to make this clearer.
No, today the General Contractors do not have a prefabrication plant, but this could obviously become the case in the future due to the strong increase of prefabrication processes in construction today.
However, changes have been made in the article to avoid confusion.
References
Bengtsson, M. (2016), “How to plan and perform a qualitative study using content analysis”, NursingPlus Open, Vol. 2, pp.8-14. doi:10.1016/j.npls.2016.01.001.
Innovation, Science and Economic Development Canada, , PRINCIPALES STATISTIQUES RELATIVES AUX PETITES ENTREPRISES, JANVIER 2019.
Nassaji, H. (2015), “Qualitative and descriptive research: Data type versus data analysis”, Language Teaching Research, Vol. 19 No. 2, pp.129-132. doi:10.1177/1362168815572747.